# Hip Active Range of Motion in Patients with Femoroacetabular Impingement Syndrome

**DOI:** 10.3390/s25041219

**Published:** 2025-02-17

**Authors:** Łukasz Stołowski, Gino Kerkhoffs, Tomasz Piontek

**Affiliations:** 1Department of Orthopedic Surgery, Rehasport Clinic, 60-201 Poznan, Poland; tomasz.piontek@rehasport.pl; 2Doctoral School, Poznan University of Medical Sciences, 60-812 Poznan, Poland; 3Department of Orthopedic Surgery, Amsterdam UMC, Academic Medical Center, Meibergdreeg 9, 1105 Amsterdam, The Netherlands; g.m.kerkhoffs@amsterdamumc.nl; 4Department of Spine Disorders and Pediatric Orthopedics, University of Medical Sciences Poznan, 61-701 Poznan, Poland

**Keywords:** femoroacetabular impingement syndrome, hip pain, range of motion, inertial measurement unit, gender differences

## Abstract

Femoroacetabular impingement syndrome (FAIS) is characterized by hip pain and restricted range of motion (ROM), typically due to structural conflict between the femoral neck and the acetabulum. This study aimed to quantify active ROM limitations in FAIS patients, comparing them with healthy controls to establish normative values, particularly in non-conflicting directions. **Methods:** A total of 53 FAIS patients scheduled for hip arthroscopy were compared to 49 healthy matched controls. Active ROM was assessed using inertial measurement unit (IMU) sensors, with measurements taken in standing and prone positions. Outcomes included flexion, external rotation, internal rotation, and total rotation ROM, alongside demographic and radiographic data. Gender-based ROM differences were also analyzed. **Results:** FAIS patients demonstrated significant reductions in flexion, internal rotation, and total rotation ROM in the involved hip, with large effect sizes, while external rotation remained unaffected. ROM in the uninvolved hip was also lower than in controls but showed increased external rotation. Gender differences were observed, with females exhibiting significantly higher internal rotation and, in healthy controls, greater total rotation than males. **Conclusions:** FAIS patients have significant active ROM restrictions in non-conflicting directions, suggesting broader joint limitations potentially tied to early hip osteoarthritis or capsular and musculoskeletal adaptations. Gender differences highlight the importance of individualized ROM assessment. This study introduces IMU-based ROM evaluation as a promising tool for diagnosing and monitoring FAIS, providing insights into functional impairments that can be used to guide targeted interventions.

## 1. Introduction

Hip pain in young and middle-aged individuals is an increasingly common reason for orthopedic consultations, with femoroacetabular impingement syndrome (FAIS) being a prevalent diagnosis. According to the Warwick consensus, FAIS is characterized by a triad of symptoms, clinical signs, and radiological findings. Key symptoms include localized pain, clicking, or stiffness, while clinical assessments involve tests such as FADDIR and FABER and accompanying restricted movement, mainly in rotation [1]. Radiological evaluations identify morphological changes, typically categorized as Cam, Pincer, or mixed types. Based on available studies, patients with FAIS exhibit significantly poorer functional outcomes, muscle strength, and range of motion (ROM) than age-matched healthy individuals [1,2].

Recent meta-analyses highlight the need for further research on FAIS, particularly in the area of active range of motion. While most analysed studies focus on the passive range of motion, it is important to consider the potential impact of active range of motion on daily activities, such as dressing and ascending stairs. Not all studies accurately define their study groups according to the FAIS criteria, particularly in terms of confirming morphological changes through imaging studies. Additionally, significant discrepancies exist in the methodologies employed, the characteristics of the study populations, and the measurement tools utilized, many of which may not be suitable for clinical settings [3].

One barrier to assessing a hip’s active range of motion is the need to stabilize the pelvis during measurement, particularly with traditional tools like goniometers and inclinometers. Inertial Measurement Unit (IMU) sensors may offer a viable solution, having demonstrated effectiveness in previous studies [4]. The device comprises a gyroscope, an accelerometer, a magnetometer, and a specialized algorithm that enables precise motion measurement. The device’s compact size and simple assembly facilitate quick and accurate measurements, particularly in one-on-one patient settings. Additionally, the sensors enable the examiner to stabilize the pelvis, helping to prevent compensatory movements. In clinical assessments of symptomatic hips, this tool has been applied to measure gait and complex movements, but not to evaluate range of motion [5,6].

More research is necessary to investigate the total rotational arc of motion in hip patients, as Louer et al. identified its restriction as a risk factor for symptomatic FAIS [7]. Examining the total rotation ROM also considers differences due to anatomical structure or acquired limitations from soft tissues [8,9]. One of the cited causes of limited range of motion in patients with FAIS is structural changes related to joint morphology. Mechanical conflict often occurs in flexion, adduction, and internal rotation [1]. Despite this, Pålsson et al. showed that a characteristic symptom for patients with FAIS might be a limitation of internal rotation in the prone position [10]. At the same time, in the mentioned meta-analysis, only a few studies assessed rotation in this position [3]. 

Before we can define limitations, it is crucial to establish what is considered normal in a given population. In various publications, there are widely differing values for the standard ranges of motion in hip joint rotation and flexion [11,12,13,14]. The results of available studies also diverge from textbook values [15]. This is often due to differences in the measurement tools used, the methodology of the study, or the group studied. Therefore, it is challenging to discuss limitations without first establishing normative values for the studied population. This underscores the critical need for normative values in our research.

Thus, the primary aim of this study is to determine the active hip-joint range of motion in healthy subjects and to compare these values with those of individuals scheduled for hip arthroscopy because of FAIS. Clinicians do not commonly assess the active range of motion in flexion while standing or in rotation while the patient is in the prone position when evaluating these patients. Therefore, it is crucial to investigate whether mobility limitations in this cohort also extend to these active ranges.

## 2. Materials and Methods

### 2.1. Participants

The study group consisted of 53 people who qualified for arthroscopy due to hip pain associated with FAIS. Recruitment for the study lasted from July 2022 to August 2024. Qualification for the procedure was determined by a single orthopedist with 25 years of experience based on history, clinical examination, and relevant imaging studies. In all patients, morphological changes consistent with FAIS and concomitant lesions were confirmed intraoperatively. Inclusion criteria were as follows: age between 18 and 60 years and referral for hip arthroscopy due to FAIS symptoms not responding to conservative treatment for a minimum of 12 weeks. Exclusion criteria were other diseases or conditions that could affect the study’s outcome.

The control group consisted of 49 healthy participants recruited for the study based on the following inclusion criteria: no history of surgeries or injuries in the lower limb or spine, no pain in the hip or spine within the last 6 months, and no other diseases that may affect the test result. To confirm normal hip function, each participant completed the hip-joint functional assessment questionnaire, a Polish version of the Hip Disability and Osteoarthritis Outcome Score (HOOS-PL). In addition, for clinical confirmation, each participant underwent the FADDIR test, which is suitable for excluding the hip joint as a source of pain [16]. The study was approved by the Bioethical Committee of the University of Medical Sciences in Poznań, Poland (no. 13/21) and met the criteria of the Declaration of Helsinki. Before the study, all participants were informed about the purpose of the study and signed consent forms indicating their willingness to participate.

### 2.2. Procedure

The study group was examined in an orthopedic clinic by a physiotherapist with 15 years of experience in examining patients with hip pain. Demographic data collected included age, gender, weight, and height. Each person completed a subjective questionnaire (HOOS PL), and their active range of motion was measured using IMU sensors (RSQ Motion, RSQ Technologies, Poznan, Poland). Radiological data and intraoperative findings were collected from the patient database.

#### 2.2.1. Assessment of Active Range of Motion

The active range of motion was assessed using a validated protocol, which showed excellent results [4]. Briefly, an IMU sensor was mounted on each participant’s limb for use in the test (Figure 1). This setup allowed unrestricted movement while simultaneously enabling the examiner to monitor compensatory movements in other body regions. The active flexion test was performed with the participant in the standing position against a wall to avoid spine movement. The sensor was mounted 5 cm above the patella, according to the protocol. Rotations were measured with the participant in the prone position, with the knee bent to 90 degrees. A foam roller with a 13 cm diameter was put between the participant’s knees for additional control of compensatory movement in the transverse plane. During the prone rotation assessment, the pelvis was stabilized by the examiner to ensure that the hip joint moved in isolation. The sensor was mounted 5 cm proximal to the ankle joint. In all directions, the movement was active. Before every measurement, each participant performed five movements to warm up and familiarize themselves with the test. The subject maintained the maximum range achieved for 1 s and recorded the measurement by pressing a button on the so-called “clicker” connected to the app. The results from the involved hips (those scheduled for surgery) and uninvolved hips from the study group and the average of the left and right hips in the healthy group were used for statistical analysis. The average result was used because the difference between the left and right hips in any direction was nonsignificant.

#### 2.2.2. Patient-Reported Outcomes

The HOOS PL personal questionnaire was used for the study. Previous research has demonstrated its utility in evaluating patients with FAIS [17]. The HOOS was selected because it provides an in-depth assessment of specific functional outcomes following hip surgery. It includes five subscales: symptoms, pain, activities of daily living (ADL), sports, and quality of life (QOL). This scale ranges from 0 to 100, with 0 indicating maximal disability and 100 indicating no symptoms. Additionally, it has been adapted and validated in Polish [18].

#### 2.2.3. Radiographic Assessment

All participants presenting with unilateral hip pain underwent a standardized set of radiographs of the hip joint, including an anteroposterior view of the pelvis and a Dunn lateral view, performed with the patients in the supine position. The alpha angle (AA) and lateral center-edge angle (LCEA) were measured by the chief orthopedic surgeon (T.P.). The AA was evaluated in the anteroposterior (AP) and Dunn lateral views as the angle formed by a line drawn from the center of the femoral head, crossing and parallel to the femoral neck, and a second line extending from the center of the femoral head to the point where the head loses its spherical shape [19]. The LCEA was defined as the angle formed between a vertical line passing through the center of the femoral head, perpendicular to the horizontal pelvic reference line, and a line extending from the center of the femoral head to the lateral edge of the acetabulum. An alpha angle ≥ 60° was used as the cutoff point for defining cam morphology, while an LCEA ≥ 40° indicated the presence of pincer morphology [20,21]. Accompanying labral or cartilage lesions were confirmed intraoperatively.

#### 2.2.4. Statistical Analysis

Statistical analyses were conducted using TIBCO Statistica 13.0 (TIBCO Software Inc., Palo Alto, CA, USA, 2017). Descriptive statistics included mean, standard deviation (SD), sample size (n), and percentage (%), along with limb-comparison differences and 95% confidence intervals (CI). Normality was tested using the Shapiro−Wilk test, and variance homogeneity was assessed using Levene’s test. Group differences were analyzed using Student’s t-test for independent samples when the normality and variance homogeneity criteria were satisfied. For normally distributed variables with unequal variances, Student’s *t*-test was applied, and the Mann−Whitney U test was used when the normality criterion was not met. Limb comparisons (involved vs. uninvolved) utilized the paired *t*-test. Effect sizes were calculated using Cohen’s d with thresholds: nonsignificant (<0.20), low (0.20–0.50), medium (0.50–0.80), and high (>0.80) [22]. Additionally, individuals for whom a small and clinically significant difference was observed were identified, and data were recorded for limb-to-limb comparisons using the formulas 0.2 × SD unoperated and 0.5 × SD unoperated [23]. The significance level was set at *p* < 0.05.

## 3. Results

### 3.1. Characteristics of Participants

Demographic and radiographic data are summarized in Table 1. All participants were matched for age, weight, height, and BMI. There was a significant difference between the study and healthy groups on the HOOS scale. Additional joint lesions were identified intraoperatively, and these patients were divided according to the required procedures. Initially, we segmented our data into two age groups (19–39 and 40–53 years) and conducted statistical comparisons for each ROM parameter. As no significant differences in ROM were observed between these cohorts (*p*-values consistently exceeding 0.05), we did not consider age-related variability in further analyses.

### 3.2. Active Range of Motion of Healthy Group Versus Study Group

Table 2 presents the differences in the active range of motion between the healthy group and the involved hip and the uninvolved hip in the study group. The study group exhibited significantly reduced flexion, internal rotation, and total rotation ROM in the involved hip compared to the healthy group, with large effect sizes. No significant difference was observed in external rotation. In the study group’s uninvolved hip, ROM was also reduced in flexion and internal rotation compared to the healthy group, with large and medium effect sizes, respectively. External rotation was significantly higher in the uninvolved hip than in the healthy group, while total rotation showed no significant reduction.

### 3.3. Involved Versus Uninvovled in Study Group

Table 3 presents the differences between the involved and uninvolved hips in the study group. The involved hip showed significantly reduced ROM in flexion and external rotation, with medium effect sizes, and in internal and total rotation, with large effect sizes. In terms of meaningful change, 60% of participants demonstrated improvement in flexion, with 42% achieving substantial improvement. In external rotation, 62% of participants showed meaningful improvement, while 51% showed substantial improvement. Internal rotation improved in the highest proportion of participants, with 83% of participants achieving meaningful improvement and 70% showing substantial improvement. For total rotation, 89% showed meaningful improvement and 77% showed substantial improvement. The thresholds for flexion were 1.6° for a small change and 4.0° for a significant change; for external rotation, these values were 1.9° and 4.8°, respectively; and for internal rotation, these values were 1.9° and 4.7°, respectively; and for total rotation, these values were 2.3° and 5.8°. These meaningful and substantial improvements highlight the clinical relevance of ROM differences, especially in internal and total rotation.

### 3.4. Gender-Specific Differences

#### 3.4.1. Gender-Based Comparison of Hip Range of Motion Across Study and Healthy Groups for Involved and Uninvolved Hips

Table 4 outlines ROM differences across flexion, external rotation, internal rotation, and total rotation between males and females in both the study and healthy groups. No significant gender differences were observed in flexion or external rotation in either group. However, females consistently demonstrated significantly higher internal rotation than males, and this trend was observed in both the study and healthy groups. Additionally, while total rotation did not differ significantly by gender in the study group, females in the healthy group exhibited a significantly greater total rotation range than males. The effect sizes highlight notable differences in hip mobility. For internal rotation, the values were 0.89 for the involved hip, 0.99 for the uninvolved hip, and 0.93 for the healthy group. Additionally, the effect size for total rotation in the healthy group was 0.83, further emphasizing the clinical relevance of these findings.

#### 3.4.2. Comparison of Hip Range of Motion Between Study and Healthy Groups by Gender

Table 5 presents ROM differences in flexion, external rotation, internal rotation, and total rotation between the study and control groups for both genders. In women, significant reductions were observed in flexion, internal rotation, and total rotation in the operated limb compared to the hips in the control group, with the effect sizes ranging from 1.13 to 2.86. Similar trends were noted for the non-operated limb, though the internal and total rotation differences did not reach statistical significance.

In men, the operated limb demonstrated significantly lower flexion, internal rotation, and total rotation than the hips in the control group, with effect sizes ranging from 1.39 to 1.58. The non-operated limb showed a significant reduction only in internal rotation, with an effect size of 0.64. External rotation showed no significant differences across groups in either gender.

## 4. Discussion

Femoroacetabular impingement syndrome (FAIS) is a complex condition commonly associated with pain and limited range of motion. A structural impingement between the femoral neck and acetabulum during combined flexion, adduction, and internal rotation is often cited as a cause. The first objective of this study was to determine whether this group also has limitations in active range of motion in directions where there is no bone conflict. The second goal was to determine the normative values for the study population in people without hip pain.

The comparison between the study and healthy groups revealed that ROM in the involved hip was significantly limited in flexion, internal rotation, and total rotation, with large effect sizes indicating substantial impairment, while external rotation remained largely unaffected. The uninvolved hip in the study group also showed reduced flexion and internal rotation compared to the healthy group, though this effect was less pronounced, and exhibited higher external rotation, with total rotation showing no significant difference. Additionally, females consistently displayed greater internal rotation across groups and higher total rotation in the healthy group, indicating a gender-related ROM advantage, especially in internal rotation. When ROM was compared within gender groups, both men and women showed significant reductions in ROM in the involved hip, with men demonstrating more significant limitations in flexion and internal rotation and women showing more pronounced deficits in total rotation.

### 4.1. What Is Normal?

Our study is the first to use IMU sensors to assess active hip ROM, a rarely studied parameter in both FAIS patients and healthy populations. Comparing existing normative values is challenging, as prior studies vary in populations, methodologies, and measurement tools. Many studies have used passive ROM assessments, typically those conducted in supine or seated positions. In our healthy group, mean internal rotation was 33.78° and external rotation was 40.29°, with the two values totaling 74.06°. Simoneau et al. assessed active rotation in a younger cohort (mean age 21.8) using a goniometer, finding slightly higher ROM values (IR: 36°, ER: 45°, Total: 81°) [11]. Ellenbecker et al., using a photographic method in healthy tennis and baseball players, reported slightly lower values (IR: 35°, ER: 35°, Total: 71°). The differences may be due to the measurement tools used and the population studied [24]. In our sample, women demonstrated a significantly greater range of internal and total rotation ROM than men, aligning with findings from a recent meta-analysis. The higher likelihood of increased femoral antetorsion and joint laxity in women may explain this difference [25].

Comparing our active hip-flexion values is challenging. We used a standing position, which limits ROM due to its active nature but involves pelvic motion that can increase values [26]. Roach et al. studied 1683 people between the ages of 25 and 70 and found 121 degrees of active hip flexion [15]. This value is higher than that found in our study, but the measurement was done in the supine position using a goniometer. Nussbaumer et al. showed that isolating the movement of the hip joint itself from the pelvis using electromagnetic sensors yielded a the range of motion that was much smaller (93.5 degrees vs. 112.1), as measured with the goniometer [26]. Notably, daily activities such as tying shoes or lifting objects require about 120° of hip flexion—a value higher than that we found, despite our participants being fully functional per the HOOS questionnaire [27]. This discrepancy may stem from our measurement method, as IMU-based results were generally lower than those obtained with a goniometer [4].

### 4.2. FAIS Patients

Our results indicate that FAIS patients scheduled for hip arthroscopy have substantial limitations in active prone rotation and standing flexion. These findings align with Albertoni et al.’s recent meta-analysis, which also found motion restrictions in FAIS patients in all directions except extension and prone external rotation [3]. Most studies focus on passive ROM data, with few using the prone position to assess rotation, as we did in our study [2,3]. Audenaert et al. compared FAIS patients (mean age 24.7) with healthy individuals and those with FAI morphology but no symptoms [28]. Their results were similar to our findings: they observed significantly reduced internal and external rotation in a prone position among FAIS patients compared to healthy and asymptomatic groups. Notably, their values were even lower than ours, possibly due to the predominance of soccer players in their sample. In one study, the authors assessed the active range of motion and found comparable limitations in rotation and flexion relative to controls, with a similar effect size to ours [29]. Comparisons are challenging, as their measurements were taken supine.

In contrast, no studies have evaluated active hip flexion in standing—an approach we believe better reflects daily movements, albeit with potential compensatory pelvic motion. Nussbaumer et al., in their study, showed that in the FAIS group, the compensatory pelvic motion allowed a significantly greater range of measurement than did isolated hip motion (84.5 vs. 103.8) [26]. In addition, previous work has shown that patients with FAIS have less pelvic mobility during flexion movement [30]. For this reason, the assessment of active flexion proposed in our work would have to be considered an assessment of global motion rather than the motion of hip joint itself.

While flexion restrictions in FAIS can be attributed to its pathomechanism and bone morphology, this does not fully explain rotational limitations in a neutral position [31]. Reduced rotation may signal early hip osteoarthritis and result from pain, joint capsule or muscle restrictions, or joint fluid—factors often present in FAIS [32,33,34,35,36]. Recently, Palson et al. demonstrated that assessing prone internal rotation and other tests can effectively identify patients with FAIS [10]. These findings highlight the importance of assessing active ROM in FAIS patients in the proposed positions. Abnormal femoral neck torsion or acetabular version, frequently seen in FAIS, may also play a role in limiting rotation [37].

For this reason, our work also considered the total rotation, which also turned out to be significantly lower in the affected hip than in the hips of participants in the healthy group and the uninvolved leg. Our work is one of the few in which this measurement was considered in this population. Tak et al. reported that limitation in total rotation may be a risk factor for groin pain [38]. In one study in whichsubjects referred for hip arthroscopy were studied, it was shown that the limitation of total rotation in the asymptomatic hip was one risk factor for future symptoms [7]. In our study group, the reduction in active range of motion also resulted in effects on the asymptomatic side compared to a healthy control group. This may be because people with FAIS have up to 78% CAM-type morphology in the asymptomatic hip, which may limit mobility [39]. Like the healthy group, women in the study group exhibited greater internal ROM. This highlights the importance of considering gender differences when assessing range of motion in patients with hip pain.

Our work is not free from limitations. We used inertial sensors, which—while adequate for measuring hip ROM—limit direct comparison with studies that used goniometers or inclinometers. Additionally, our choice of test positions, such as standing active flexion, was intended to reflect daily activities but mean that our results are not directly comparable with those of prior studies. As this is the first study of its kind, results could be compared only to those of the healthy group and those of the asymptomatic leg. Finally, the same investigators conducted all assessments with awareness of the affected hip, which may have introduced bias. In addition, participants were pre-selected for hip arthroscopy and therefore were likely experiencing more pain that the broader FAIS population, which could influence ROM measurements and limit generalizability to the broader FAIS population. Patients had also undergone various conservative treatments prior to surgery, potentially affecting ROM outcomes. The main strength of this study is its well-defined FAIS cohort, with membership confirmed through clinical assessment, symptoms, imaging, and intraoperative findings. Our IMU-based protocol for assessing active ROM is reproducible and easy to implement in clinical settings, and it uses a validated device compatible with inclinometer and goniometer measurements. This is one of the few studies in which inertial sensors have been used to evaluate patients with a specific disease entity and the first such study to assess an FAIS cohort. Notably, the ROM limitations in FAIS patients were both statistically and clinically significant, highlighting the practical value of assessing active ROM in FAIS patients in future research and clinical practice.

## 5. Conclusions

Femoroacetabular impingement syndrome is an increasingly well-recognized clinical entity that is frequently associated with restricted range of motion, particularly in flexion, adduction, and internal rotation. Our study provides novel insights, demonstrating that significant ROM limitations exist in less obvious directions, including total rotation, particularly on the symptomatic side. Importantly, our findings indicate that women exhibit greater internal and total rotation than men, both in the FAIS group and in healthy individuals, underlining the need to account for gender differences in clinical assessments.

This study introduces an innovative method utilizing IMU sensors for assessing active hip-joint ROM that may better reflect functional movements encountered in daily activities. While our findings are statistically and clinically significant, they represent a first step toward defining normative values for FAIS patients using this approach. Future studies should validate the observed patterns in larger cohorts and investigate their clinical implications over time. Such efforts will further establish the reliability of this method and its potential for diagnosing and monitoring hip-related conditions.

## Figures and Tables

**Figure 1 sensors-25-01219-f001:**
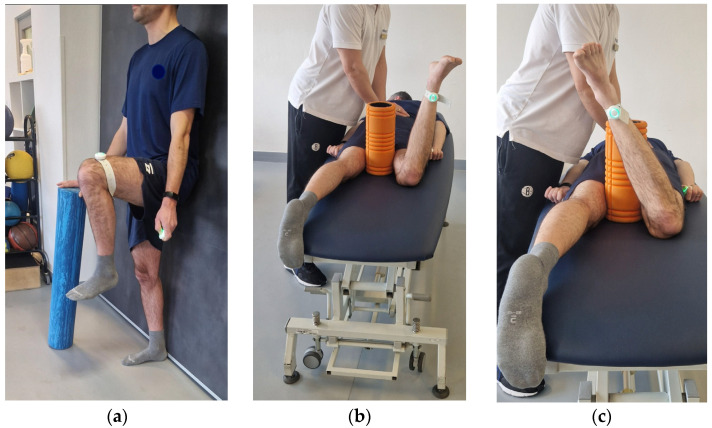
Range-of-motion assessment with IMU (**a**) active flexion, (**b**) active internal rotation, (**c**) active external rotation.

**Table 1 sensors-25-01219-t001:** General characteristics of participants.

Parameters	Study Group (*n* = 53)	Healthy Group (*n* = 49)	*p*-Value
Sex/men [% (*n*)]	68 (36)	67 (33)	0.950 ^c^
	Mean ± SD		
Age [years]	36.3 ± 9.3	35.4 ± 8.0	0.613 ^a^
Weight [kg]	76.6 ± 13.7	75.5 ± 13.0	0.804 ^a^
Height [cm]	177.0 ± 8.1	176.2 ± 9.5	0.338 ^a^
BMI [kg/m^2^]	24.3 ± 3.3	24.1 ± 2.7	0.923 ^b^
HOOS [pts.]	58.2 ± 14.9	99.4 ± 1.1	<0.001 ^b^*
LCEA [°]	35.6 ± 5.4	n/a	n/a
AA AP [°]	67.3 ± 12.5	n/a	n/a
AA Dunn [°]	61.4 ± 10.6	n/a	n/a
Procedure concomitant with osteoplasty [% (*n*)]			
Labrum repair	54.7 (29)	n/a	n/a
Labrum reconstruction	7.5 (4)	n/a	n/a
Labrum resection	13.2 (7)	n/a	n/a
Microfracture	13.2 (7)	n/a	n/a

^a^ Student’s *t*-test for independent samples, ^b^ Mann−Whitney U test, ^c^ Pearson’s chi-squared test. * Significance level. Abbrevations: BMI = Body mass index; HOOS = Hip Disability and Osteoarthritis Score; pts = points; LCEA = Lateral Center Edge Angle; AA AP = Alpha Angle Anterior Posterior View; AA Dunn = Alpha angle Dunn view; n/a = Not applicable.

**Table 2 sensors-25-01219-t002:** Active range of motion of healthy group versus study group.

ROM (°)	Healthy Group *n* = 49	Involved Leg *n* = 53	*p*-Value	Effect Size	Uninvolved Leg *n* = 53	*p*-Value	Effect Size
	Mean ± SD				Mean ± SD		
Flexion	108.2 ± 5.5	99.0 ± 8.8	<0.001 ^b^*	1.30	102.4 ± 8.1	0.001 ^b^*	0.83
External rotation	40.2 ± 8.4	38 ± 10.5	0.177 ^b^	0.23	44.1 ± 9.5	0.040 ^b^*	0.43
Internal rotation	33.7 ± 9.9	18.5 ± 11.7	<0.001 ^a^*	1.40	27.9 ± 9.4	0.003 ^a^*	0.60
Total Rotation	73.9 ± 8.7	56.5 ± 14.1	<0.001 ^c^*	1.47	71.9 ± 11.6	0.358 ^a^	0.18

^a^ Student’s *t*-test for independent samples, ^b^ Mann-Whitney U test, ^c^ test with independent estimate variance. * Significance level.

**Table 3 sensors-25-01219-t003:** Active range of motion of the involved versus uninvolved hip in the study group.

ROM (°)	Uninvolved Hip	Involved Hip	Difference	95% CI Difference	*p*-Value	Effect Size	Proportion Improvement	Proportion Substantial Improvement
	Mean ± SD						% (*n*)	
Flexion	102.4 ± 8.1	98.6 ± 8.8	3.8	2.1–5.5	<0.001 ^a^*	0.59	60 (32)	42 (22)
External rotation	44.1 ± 9.5	38.0 ± 10.5	6.1	3.4–8.7	<0.001 ^a^*	0.64	62 (33)	51 (27)
Internal rotation	27.9 ± 9.4	18.5 ± 11.7	9.4	7.1–11.7	<0.001 ^a^*	1.02	83 (44)	70 (37)
Total rotation	71.9 ± 11.6	56.5 ± 14.1	15.5	12.3–18.6	<0.001 ^a^*	1.30	89 (47)	77 (41)

^a^ Student’s *t*-test for dependent samples. * Significance level.

**Table 4 sensors-25-01219-t004:** Gender-based comparison of hip range of motion across study and healthy groups for involved and uninvolved hips.

ROM (°)	Study Group						Healthy Group		
	Involved Hip			Uninvolved Hip					
	Male *n* = 36	Female *n* = 17	*p*-Value	Male *n* = 36	Female *n* = 17	*p*-Value	Male *n* = 33	Female *n* = 16	*p*-Value
	Mean ± SD			Mean ± SD			Mean ± SD		
Flexion	97.7 ± 9.1	100.5 ± 7.9	0.272 ^a^	101.5 ± 9.4	104.3 ± 4.1	0.244 ^b^	107.6 ± 4.2	109.3 ± 7.5	0.433 ^c^
External rotation	38.9 ± 9.2	36.1 ± 12.9	0.359 ^a^	45.3 ± 9.6	41.5 ± 9.2	0.185 ^a^	40.8 ± 9.1	38.9 ± 6.8	0.478 ^a^
Internal rotation	15.4 ± 10.3	25.1 ± 11.9	0.004 ^a^*	25.2 ± 8.6	33.7 ± 8.4	0.002 ^a^*	30.9± 9.2	39.4 ± 8.9	0.004 ^a^*
Total rotation	54.3 ± 14.4	61.1 ± 12.5	0.103 ^a^	70.5 ± 12.8	75.9± 8.1	0.171 ^a^	71.7 ± 7.6	78.4 ± 9.5	0.038 ^b^*

^a^ Student’s *t*-test for dependent samples, ^b^ Mann−Whitney U test, ^c^ test with independent estimate variance. * Significance level.

**Table 5 sensors-25-01219-t005:** Comparison of hip ROM between study and healthy groups by gender.

Sex	ROM (°)	Healthy Group	Study Group, Involved	*p*-Value	Effect Size	Study Group, Uninvolved	*p*-Value	Effect Size
Women	FX	109.25 ± 7.53	100.53 ± 7.88	0.003 ^a^*	1.13	104.3 ± 4.05	0.029 ^c^*	1.16
	ER	38.97 ± 6.88	36.06 ± 12.89	0.423 ^c^	0.28	41.53 ± 9.17	0.373 ^a^	0.31
	IR	39.38 ± 8.88	25.06 ± 11.94	0.004 ^a^*	1.36	33.65 ± 8.43	0.067 ^a^	0.66
	TROT	78.35 ± 9.44	61.12 ± 12.52	<0.001 ^b^*	2.86	75.18 ± 8.06	0.456 ^b^	0.38
Men	FX	107.64 ± 4.16	97.67 ± 9.14	<0.001 ^c^*	1.39	101.5 ± 9.34	0.058 ^b^	1.07
	ER	40.82 ± 9.14	38.93 ± 9.25	0.395 ^a^	0.20	45.28 ± 9.61	0.053 ^a^	0.48
	IR	30.88 ± 9.21	15.42 ± 10.30	<0.001 ^a^*	1.58	25.2 ± 8.62	0.010 ^a^*	0.64
	TROT	71.70± 7.56	54.34 ± 14.43	<0.001 ^c^*	1.49	70.47 ± 12.79	0.625 ^c^	0.12

^a^ Student’s *t*-test for dependent samples, ^b^ Mann−Whitney U test, ^c^ test with independent estimate variance. * Significance level. Abbrevations: Fx = active flexion, ER = Active external rotation, IR = active internal rotation, TROT = active total rotation.

## Data Availability

The data presented in this study are available from the corresponding author upon reasonable request due to privacy considerations.

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
