# Peer review of "Hip Active Range of Motion in Patients with Femoroacetabular Impingement Syndrome"

_sensors, 2025, doi:10.3390/s25041219_

Round 1
Reviewer 1 Report
Comments and Suggestions for Authors
This paper studies femoroacetabular impingement syndrome problem by using inertial measurement unit, and analyzed results of 53 FAIS patients. The research has certain guiding significance for the intervention of FAIS patients. However, the article still has the following problems that need to be modified.
1. The innovation of the article is not outstanding enough. Is the innovation just using IMU to measure ROM?
2. Please explain the definition of 'involved' and 'uninvolved'.
3. The major problem of the article is that only some results are obtained by IMU measurement method, but the rules or phenomena found are not verified. In other words, if the author can summarize the rule of the test results and give specific values, and then carry out several groups of tests, it will be more convincing if the conclusions can be verified.
4. There is an extra full stop on line 22 of the article abstract.
5. It is better to give a schematic diagram of the scenario of measuring ROM with IMU.
Author Response
We thank you very kindly for all your comments and suggestions. They helped improve our article a lot. For clarity, we have highlighted the changes made to the manuscript in blue. Below are our responses:
Comments 1. The innovation of the article is not outstanding enough. Is the innovation just using IMU to measure ROM?
Response 1:
Thank you for the question. We acknowledge that examining ROM alone may not fully showcase the IMU's capabilities. However, in our opinion this study is significant because:
- It is the first to use IMUs for assessing FAIS patients and one of the few studies focusing on the hip joint.
- It examines active ROM, which is rarely studied, even in other anatomical regions.
- It explores movement directions in FAIS patients that have not been previously studied (e.g., active standing flexion) or are seldom investigated (e.g., active prone rotation).
The use of a single sensor enables free pelvic stabilization and measurement, offering advantages over other clinical tools.
Comments 2: Please explain the definition of 'involved' and 'uninvolved'.
Response 2:
Thank you for pointing this out. We compared patients scheduled for arthroscopy (study group) to healthy volunteers. In the study group, the term 'involved hip' refers to the hip scheduled for arthroscopy, while the term 'uninvolved hip' refers to the contralateral hip of the same patient. To clarify this, we added an explanation in parentheses in line 135 of the manuscript.
Comments 3: The major problem of the article is that only some results are obtained by IMU measurement method, but the rules or phenomena found are not verified. In other words, if the author can summarize the rule of the test results and give specific values, and then carry out several groups of tests, it will be more convincing if the conclusions can be verified.
Response 3:
We thank the reviewer for their insightful comment regarding verifying the observed patterns in our study. As this is the first study to use IMU sensors to assess active ROM in FAIS patients, our primary aim was exploratory—to identify significant ROM limitations and gender-specific differences compared to healthy controls. While our findings revealed consistent trends, such as reduced internal and total rotation in FAIS patients and gender differences favoring greater ROM in women, we agree that further validation is necessary. To address this, we have revised the Conclusion sections to highlight the study's exploratory nature and recommend future research using larger, more diverse cohorts and longitudinal designs to confirm these results.
Comments 4: There is an extra full stop on line 22 of the article abstract.
Response 4:
Thank you. We corrected it.
Comments 5: It is better to give a schematic diagram of the scenario of measuring ROM with IMU.
Response 5:
We appreciate the reviewer’s suggestion to include a schematic diagram of the measurement scenario. In response, we have added a photograph (Figure 1) illustrating the active range of motion tests, including flexion, internal rotation, and external rotation, with the IMU sensors mounted according to the validated protocol. This protocol, which has been validated in a previous study, ensures accuracy and reliability in the assessment(DOI:3390/s23218782). Additionally, we clarified the use of pelvis stabilization during prone rotation assessments in the Methods section to highlight measures taken to ensure proper isolation of hip joint movement. We believe that the inclusion of the photograph, combined with the validated protocol and detailed description, sufficiently addresses the reviewer’s suggestion and provides a comprehensive understanding of the measurement setup
Reviewer 2 Report
Comments and Suggestions for Authors
I reviewed the manuscript entitled "Hip Active Range of Motion in Patients with Femoroacetabular Impingement Syndrome" by Lukasz Stolowski et al. I found manuscript interesting and would be good fit for sensors journal. I would be happy to accept the manuscript for publication after the minor corrections listed below.
- It would be beneficial to compare gender-specific changes between the healthy group and the study group. Therefore, I suggest including a table or graph that compares the data for healthy males versus study group males and healthy females versus study group females.
- The age of the patients ranges from 18 to 60 years. I believe this age group is too broad, as young adults may exhibit significant differences in range of motion (ROM). Therefore, it would be beneficial to segment the data into specific cohorts, such as 18-30, 30-45, and 45-60. You can create cohorts based on your data. It would be worth mentioning in the manuscript if there is no age-related difference.
- It would be great to include key results in the figure format.
Author Response
We thank you very kindly for all your comments and suggestions. They helped improve our article a lot. For clarity, we have highlighted the changes made to the manuscript in blue. Below are our responses:
Comments 1: It would be beneficial to compare gender-specific changes between the healthy group and the study group. Therefore, I suggest including a table or graph that compares the data for healthy males versus study group males and healthy females versus study group females.
Response 1:
We appreciate the reviewer’s suggestion and agree that gender-specific comparisons provide valuable insights into ROM impairments in FAIS. To address this, we have included a dedicated section in the manuscript (Section 3.4.2) and Table 5, which presents ROM differences between the study and control groups for males and females separately. Our findings indicate that both genders exhibit significant flexion, internal rotation, and total rotation reductions in the involved hip, with some differences in effect sizes and trends. Women demonstrated more pronounced total rotation deficits, whereas men showed greater flexion and internal rotation limitations.
Comments 2: The age of the patients ranges from 18 to 60 years. I believe this age group is too broad, as young adults may exhibit significant differences in range of motion (ROM). Therefore, it would be beneficial to segment the data into specific cohorts, such as 18-30, 30-45, and 45-60. You can create cohorts based on your data. It would be worth mentioning in the manuscript if there is no age-related difference.
Response 2:
We agree that age-related differences in ROM could be relevant, which is why we conducted an analysis based on age. We segmented our data into two age groups (19–39 and 40–53 years) and performed statistical comparisons for each ROM parameter. The analysis revealed no significant differences between these cohorts, with p-values consistently exceeding 0.05. This suggests that within this age range, age-related differences do not meaningfully impact active ROM in our sample population. Therefore, further subdivision into narrower cohorts (e.g., 18–30, 30–45, 45–60) would likely yield similar findings due to the absence of a significant trend in our data.
Comments 3: It would be great to include key results in the figure format.
Response 3:
We appreciate the suggestion to include key results in figure format to enhance clarity. However, we believe that the current presentation of the data in tables provides comprehensive and detailed information, including mean values, standard deviations, and statistical significance for all comparisons. Presenting the same information in figures would likely lead to redundancy and potentially omit the granular details that are critical for interpretation.
Additionally, we have structured the tables to allow for straightforward comparison between groups and variables, ensuring transparency and ease of analysis for readers. For these reasons, we kindly request to retain the tables as they are, while ensuring the discussion and interpretation of the results are clear and accessible within the text.